# Perceptual and semantic maps in individual humans share structural features that predict creative abilities
Johannes P.-H. Seiler [1,4] ✉, Jonas Elpelt [2,3,4], Aida Ghobadi[1], Matthias Kaschube [2,3] & Simon Rumpel [1] ✉

Building perceptual and associative links between internal representations is a fundamental neural process, allowing individuals to structure their knowledge about the world and combine it to enable efficient and creative behavior. In this context, the representational similarity between pairs of represented entities is thought to reflect their associative linkage at different levels of sensory processing, ranging from lower-order perceptual levels up to higher-order semantic levels. While recently specific structural features of semantic representational maps were linked with creative abilities of individual humans, it remains unclear if these features are also shared on lower level, perceptual maps. Here, we address this question by presenting 148 human participants with psychophysical scaling tasks, using two sets of independent and qualitatively distinct stimuli, to probe representational map structures in the lower-order auditory and the higher-order semantic domain. We quantify individual representational features with graph-theoretical measures and demonstrate a robust correlation of representational structures in the perceptual auditory and semantic modality. We delineate these shared representational features to predict multiple verbal standard measures of creativity, observing that both, semantic and auditory features, reflect creative abilities. Our findings indicate that the general, modality-overarching representational geometry of an individual is a relevant underpinning of creative thought.

Our complex world with its ever-changing environmental conditions challenges individuals to integrate the information of different sensory stimuli in order to effectively inform behavior. The perception of sensory stimuli is inevitably embedded in larger representational networks, setting an incoming stimulus in relation to previous experience and internal priors[1–4]. These complex relational structures between different internally represented elements of information have been conceptualized as *representational maps*[2,5–8]. On a representational map, different representational entities, e.g. sensory stimuli, are tied to each other in a relational space reflecting their perceptual and associative linkage[4,8–11]. Classically, a representational map is estimated by assessing pairwise similarities between the psychometric or neuronal representations of a diverse, rich set of sensory stimuli, and then reducing this high-dimensional abstract space to a lower number of interpretable dimensions[6,8,12–14]. Importantly, a particular sensory stimulus might at the same time be represented on several representational maps along the processing hierarchy which qualitatively differ in what

properties are encoded as representational similarity[8,15]. For instance, the representations of two phonetically similar spoken words, such as "car" and "tar", might be embedded on a semantic, meaning-related level at a much higher distance, differing strongly from its embedding on a purely perceptual, phonetic level. Importantly, also pairs of words that are phonetically highly different may be embedded in high proximity on a semantic representational map when their meanings are largely overlapping. In line with this perspective, recent efforts have emphasized multilayered and multiplex knowledge networks to account for different types of information, conveyed by the structure of a complex representational map at the same time[16]. This framework assumes that the combination of multiple maps, tuned to represent different, complementary features of the same informational entities, yields a higher-order framework, depicting the relations by which represented information affects cognition.

Interestingly, recent studies indicate that the representational structure of verbal stimuli at a high, semantic level, reflects the ability of an individual

[1]Institute of Physiology, Focus Program Translational Neurosciences, University Medical Center of the Johannes Gutenberg University Mainz, Mainz, Germany. [2]Frankfurt Institute for Advanced Studies, Frankfurt am Main, Germany. [3]Institute of Computer Science, Goethe University Frankfurt, Frankfurt am Main, Germany. [4]These authors contributed equally: Johannes P.-H. Seiler, Jonas Elpelt. ✉e-mail: johseile@uni-mainz.de; sirumpel@uni-mainz.de

to form creative associations[17–19]. These findings leverage classical associative concepts of creativity, characterizing the creative process by the ability to build unconventional and useful associations within and across different cognitive domains[20–22]. While current research on creativity is largely confined to human studies utilizing language to assess creative abilities[20,23], the recently suggested linkage of creativity with basic representational features indicates that creativity could also be assessed on a more fundamental perceptual level[24]. In this context, pareidolic perceptions, i.e. the notion of objects in ambiguous stimuli, have been linked with higher creativity[25]. Moreover, synesthesia – the trait to associate stimuli across different sensory modalities – has been linked with higher creative achievement[26,27], supporting the idea that creativity is associated with higher tendencies to form perceptual linkages[24,25]. Thus, assessing the representational characteristics of an individual on perceptual and semantic level features could allow us to also gain insights about its creative ability.

While multiple factors such as emotional states or executive processes can potentially affect representational maps and their readout[8,28], it was demonstrated that straightforward psychophysical estimates of representational similarity (e.g. similarity ratings of stimuli) show consistent structure with neurometric measurements of representational similarity (e.g. electrophysiological brain recordings)[29], even generalizing across different species[30,31]. Hence, psychophysical assessments of representational similarity allow uncomplicated, valid assessments of individual representational maps in different domains, such as perception and semantics.

In this study, we test the hypothesis that specific structural features of representational maps are shared across different modalities in an individual, accounting for general cognitive properties and hence shaping creative ability. We address this by conducting psychophysical scaling tasks in healthy humans to measure their representational structures of (i) the higher-order semantics of visually presented word stimuli, and (ii) lower-order perceptual properties of auditory presented arbitrary pulsed sound stimuli. We applied graph-theoretical measures to parametrically describe the structural features of individual semantic and perceptual maps, finding that specific features of representational maps are robustly correlated across the auditory and semantic modality. Moreover, we use statistical modeling and find that the shared structural features of representational maps serve as significant predictors for standard metrics of creativity. Together, our findings indicate a common representational architecture for each individual that joints perception and association across different modalities, thus shaping creative thought and behavior.

## Methods
The study was approved by the local ethics committee (Ethikkommission der Landesärztekammer Rheinland-Pfalz, processing number 2024-17477). There was no pre-registration for the study. Written informed consent was obtained from all participants of the study.

### Study cohort
A total number of 162 healthy students from the University of Mainz were recruited to participate in our study, using an online recruiting system[32]. Exclusion criteria were active psychiatric or neurological disorders as well as insufficient German language skills, both assessed by self-reports (see general information questionnaire below). From the initial sample, 14 participants encountered technical problems during the experiment, leading to incomplete data and exclusion from our analyses. Thus, a final sample of 148 participants was used for the analyses of this study. The sample comprised 116 women (78.4%) and 32 men (21.6%), with an average age of 22.7 years (for further demographic information see Supplementary Table 1). Gender and age information was assessed through self-reports.

### Experimental procedure
The experiment of the study was conducted on a single day in the facilities of the Mainz Behavioral and Experimental Laboratory (MABELLA). Participants were introduced to the experiment and asked for their agreement to participate. After providing consent, participants started to work on the experiment, implemented in a custom MATLAB® program which was presented on a standard computer screen. For the duration of the experiment, participants were instructed to wear headphones, which were also used to deliver the auditory stimuli. The study comprised different steps listed in the following (see Fig. 1A).

**Psychometric questionnaires.** First, participants were asked to fill out a battery of different self-report scales in order to assess demographic and psychometric properties, such as personality traits and properties of mental health and resilience. Specifically, participants reported *general information* (gender, age, weight, size, patient history), Big Five personality characteristics (*BFI-10*: Big Five Inventory[33] covering neuroticism, extraversion, openness, agreeableness, conscientiousness), trait anxiety (*STAI-Y*: State Trait Anxiety Inventory[34]), indicators of mental resilience (*BRS*: Brief Resilience Scale[35,36]), boredom proneness (*BPS*: Boredom Proneness Scale[37,38]) and state boredom at the start of the experiment (*MSBS*: Multidimensional State Boredom Scale[38,39]). All questionnaires were presented in German language.

**Verbal assessments of creative associations.** In the next step, participants completed a set of language-based association tasks to assess divergent thinking as a proxy for individual creativity[20,40]:

Alternative uses task[41] (*AUT*). Participants were asked to list potential uses for a glass bottle, potential uses for a knife, things that can be round, and things that can cause noise. For each of the four questions, the response time was limited to three minutes. To estimate individual fluency and originality in creativity, we assessed the number and variety of submitted answers. Specifically, we measured the mean maximal semantic distance to the anchor word in all questions (see "Methods" section), yielding a readout of divergent thinking[42].

Compound remote associations task[21] (*C-RAT*). Participants were asked to find a target word building a compound with three given anchor words. The number of correct compounds then served as a proxy for individuals' creative ability. The response time was limited to one minute per item. We included a set of 10 items which have been validated in a previous study to cover a broad spectrum of difficulty, concerning the fraction of participants able to solve them and the time needed to find a solution[43] (items in German with the correct answer: Wasser-Speise-Jod -> Salz, Hals-Fahrrad-Perle -> Kette, Ruhe-Sommer-Kaffee -> Pause, Pelz-Tasche-Schutz -> Mantel, Anzug-Kapsel-Welt -> Raum, Natur-Tüte-Welt -> Wunder, Hut-Wurst-Ring -> Finger, Gruppe-Druck-Gefäß -> Blut, Hals-Kampf-Hilfe -> Schrei, Schmerz-Fell-Tanz -> Bauch).

Divergent associations task[44] (*DAT*). In this simple task, participants are asked to list up to ten words that are as dissimilar as possible in all their meanings and use cases. Divergent thinking in this task is estimated as a proxy for creativity by quantifying the mean semantic distance between the first seven valid responses (see "Methods" section).

**Auditory and semantic scaling task.** In the following step, we presented participants with a task to scale the subjectively perceived similarities between different stimuli. This task was conducted in two independent blocks, the first one covering auditory stimuli and the second one covering semantic stimuli. For the auditory scaling task, we used a set of 20 different pulsed auditory stimuli, each having a length of 1 s, but varying in the number and temporal pattern of short white noise pulses distributed over the stimulus (single pulse duration: 30 ms with a linear up- and down-ramp of 5 ms; the temporal pattern of pulses was randomized with a constrained minimal interval of 50 ms between consecutive pulses; for envelopes of the stimulus set see Fig. 1B, C; sound stimuli are available via the link in the data availability statement). The semantic stimulus set consisted of 20 German words from five different categories, which were presented to the participants visually on the computer screen (see

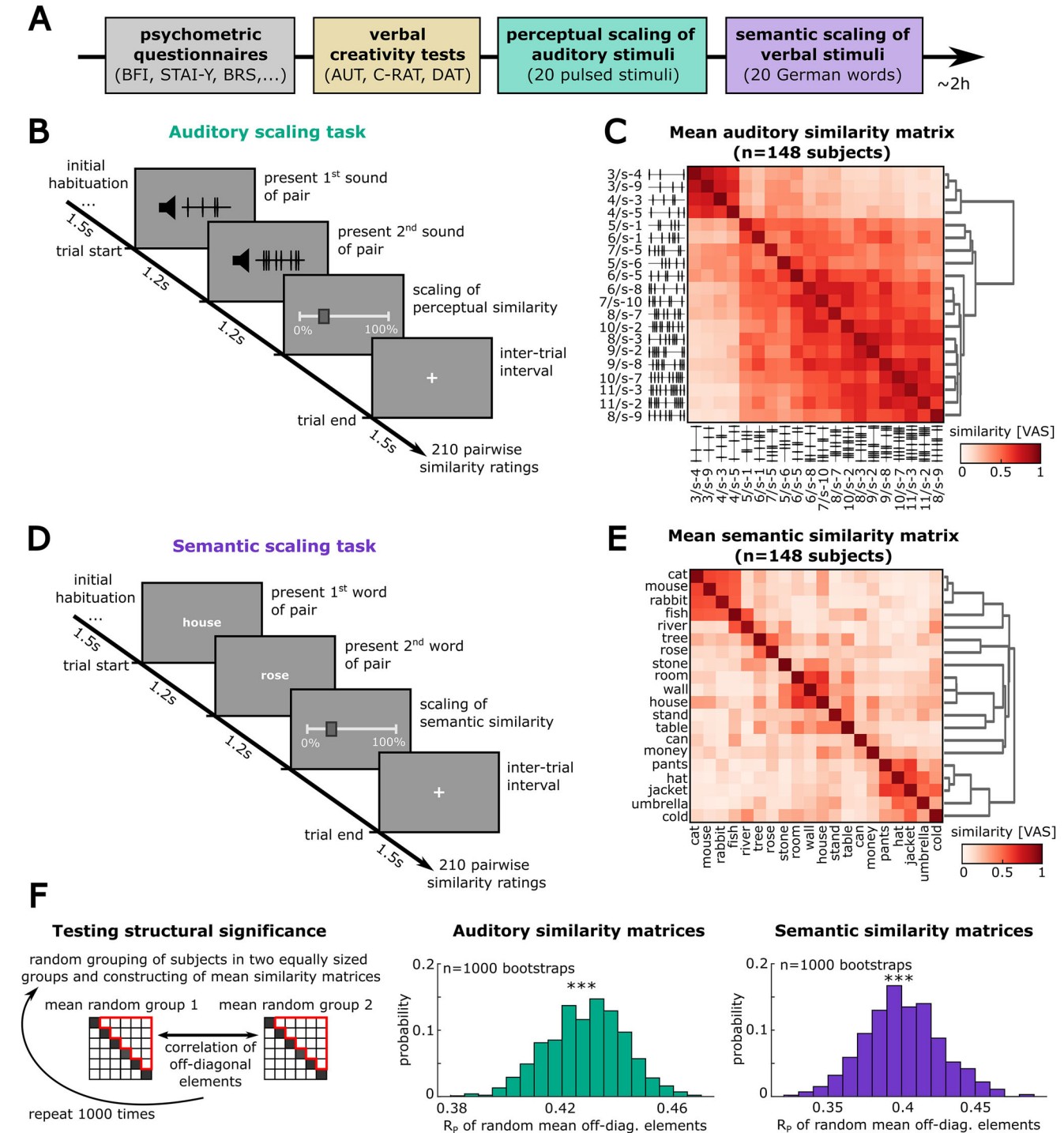

**Fig. 1 | Assessing semantic and perceptual representational similarities with a scaling task. A** Flow of experiment ($n = 148$ participants; see "Methods" section). **B** Participants were initially presented twice with all possible stimuli to habituate them to the stimulus set. Then, in each trial of the task, two pulsed auditory stimuli were presented with the subsequent request to rate their similarity (see "Methods" section). **C** Matrix of the mean auditory similarity ratings over all participants and stimulus pairs (VAS: visual analog scale; the stimulus labels indicate the number of pulses per second and an arbitrary suffix for identification). The stimuli are sorted by the distance of their similarity patterns (dendrogram on the right, "Methods" section). **D, E** Equivalent semantic scaling task with word stimuli and the corresponding mean similarity matrix. Here, participants were specifically instructed to rate the semantic similarity of the word pairs. **F** Bootstrap test to quantify the statistical reliability of the off-diagonal patterns in the mean similarity matrices. For this, the similarity matrices of all participants were randomly assigned to two evenly-sized groups ($n = 74$ matrices per group), before building the mean for each group and correlating their average similarity patterns as flattened vectors (see "Methods" section). This procedure was repeated yielding a distribution of correlation values that indicate the reliability of the observed off-diagonal similarity patterns across individuals (***$p < 0.001$ in a $t$-test against zero).

Fig. 1D, E; modified from a previous study investigating effects of semantic network features on creativity[18]; category "animals": rabbit, cat, mouse, fish; category "natural objects": tree, river, rose, stone; category "building-related objects": room, house, wall, stand; category "clothing":

pants, hat, jacket, umbrella; category "diverse objects": table, can, money, cold; German version of the words: Hase, Katze, Maus, Fisch, Baum, Fluss, Rose, Stein, Raum, Haus, Wand, Stand, Hose, Hut, Jacke, Schirm, Tisch, Dose, Geld, Kälte). Each block of the scaling task, auditory or

semantic, started with an initial presentation of all stimuli of the stimulus set (twice repeating the 20 stimuli in random order) to familiarize participants with the stimulus set and its variety. The interval for stimulus presentation was fixed to 1.2 s with an inter-trial interval of 1.5 s. Next, random pairs of stimuli were consecutively presented to the participants with an interval of 1.5 s in between. Then, a slider was displayed on the screen with the request to rate the similarity between both presented stimuli on a visual analog scale (VAS) from "not similar at all" to "completely identical". For the semantic stimuli, we explicitly asked participants to rate the semantic similarity between both words. Participants had no time limit for moving the slider and rating the similarity (average response times pooled for both modalities were homogenous across participants: mean ± SD 2.47 s ± 0.59 s).

After moving the slider, participants were able to submit their responses, which led to the start of the next trial by presenting another pair of sounds after an inter-trial interval of 1.5 s. For each trial, the slider was initialized with the intermediate neutral value and had to be actively moved in order to enable the submission of the response. For each block of the scaling task with 20 stimuli per modality, this resulted in 210 trials for each participant to cover all pairs of stimuli, including pairs of identical stimuli that were rated. The order to present the pairs of stimuli and the order of presenting the stimuli within each trial were randomized.

**Additional elements of the experiment independent of this study.**
Between the auditory and verbal scaling task, participants performed three rounds of a two-alternative forced choice task, where they had to discriminate the set of pulsed auditory stimuli according to different reinforcement contingencies. Moreover, participants reported their state of boredom and the assumed rule after each round of the discrimination task. After completing the semantic scaling task, participants also filled out some questions about their perception and sentiment of five short texts. These texts were presented after completing the semantic scaling task in order to prevent interference with the similarity ratings. The data of these subparts is not considered in the current study but is used for independent studies on text perception and auditory perception (currently in preparation for publication).

**Statistical analysis**
All analyses were conducted using the MATLAB® statistics and machine learning toolbox (The Mathworks Inc., Natick, Massachusetts, USA, version R2022a) and Python (version 3.8.5; packages: network, scipy, scikit-learn, SentenceTransformers). All data was pseudonymized before analyzing.

**Psychometric questionnaires.** The self-reported data was analyzed by computing the sum score for each questionnaire. Participants who accidentally skipped single items of a questionnaire or subscale were excluded from the respective analysis. This exclusion explains deviations from the total number of recruited participants and the reported $n$ of the respective analysis.

**Analysis of verbal creativity assessments based on semantic distance.** We assessed individual abilities to build creative verbal associations by quantifying the semantic distance of each participant's responses[45,46] in the AUT and DAT, using a state-of-the-art open-source multilingual sentence embedding model based on RoBERTa architecture (https://huggingface.co/T-Systems-onsite/cross-en-de-roberta-sentence-transformer). For the AUT, each submitted answer was embedded sentence-wise quantifying the maximal semantic distance to the question[42]. We compared the resulting semantic distance scores with ratings of originality based on the OCSAI method[47], an alternative automated scoring technique for the AUT based on large-language models, finding a significant correlation between both metrics (Supplementary Fig. 8A), and qualitatively equivalent results when regressing individual OCSAI scores by representational map features (Supplementary Fig. 8B). For the DAT, we quantified the mean crosswise

semantic distance between the first seven valid submitted words. Moreover, we computed the mean number of submitted responses in the AUT, yielding a measure of fluency in the association process. The C-RAT was evaluated by summing up the number of correct answers from all ten items. Participants who did not submit any valid (i.e. syntactically coherent) response to the questions were excluded from the respective analysis. This explains deviations in the reported sample size for the respective analysis.

**Testing the reliability and validity of the auditory and semantic similarity ratings.** Based on the 210 pairwise similarity ratings of each participant for the auditory and semantic stimulus sets, we constructed a similarity matrix $M$ for each participant and modality respectively. Building the mean similarity matrix over all participants yields a global estimate of stimulus similarities for both modalities separately. To test the reliability of these estimations across participants, we conducted a bootstrap test, where we randomly separated our dataset into two equally sized groups ($n = 74$ matrices per group) and built the mean similarity matrix for each of the groups. Then, we assessed the structural similarity of both mean similarity matrices by computing the Pearson correlation of the unique off-diagonal elements, flattened into a vector. This yields a numerical value ranging from $-1$ to 1, where 0 would indicate no similarity of the mean matrices, 1 would indicate completely congruent matrices and $-1$ would indicate inverted matrices. We repeated this procedure 1000 times for both modalities independently, resulting in a distribution of correlation coefficients, found to be on average positive in both modalities (Fig. 1F, mean ± SEM: auditory 0.428 ± 0.014, semantic 0.401 ± 0.026; both $p < 0.001$ in Wilcoxon signed rank test against a median of 0). This indicates that perceptual relations within each modality are significantly shared across individuals.

To corroborate this validation for the semantic ratings, we compared the mean semantic similarity matrix from our experiment against an analogous matrix of reference similarities obtained from a standard word embedding model (Supplementary Fig. 1A, B). Here, we found a significant correlation between both matrices (Supplementary Fig. 1C), indicating that the semantic ratings of our study match general semantic relations of German-speaking populations, estimated from a non-experimental setting. For the presentation of the similarity matrices, we sorted the stimuli according to their relative similarity pattern, using the dendrogram obtained from hierarchical clustering of the matrices (specifying the shortest Euclidean distance as a distance algorithm).

**Transformation of similarity ratings into a similarity matrix and graph.** Based on the individual similarity ratings obtained from the visual analog scales in a given modality, we constructed a similarity matrix and a corresponding graph, providing estimates of an individual's representational map, and capturing the perceptual relations between all the stimuli in the respective modality. For each participant, based on the similarity ratings across all 210 stimulus pairs, we constructed the similarity matrix $M$. This matrix $M$ contained the raw similarity ratings from the respective modality, expressed as numerical values between 0 (no/minimal similarity) and 1 (maximal similarity). In order to quantify the structure of an individual map, we used the matrix $M$ as an adjacency matrix to construct a corresponding fully connected graph $G$ with each stimulus as a node and the pairwise similarity ratings between stimuli as edges. In accordance with the similarity values, the edge weights thus range between values of 0 and 1. In the next step, we used $M$ and $G$ as quantitative proxies of individual representational maps in a given modality, and computed a broad set of features to describe their strcture (referred to as *map features*). All map features are listed in the following.

**Graph and map features to estimate the structure of individual perceptual and semantic maps**
Mean off-diagonal similarity. The mean value of all 190 unique off-diagonal pairwise similarity ratings.

**Intra-categorical similarity.** The mean value of all edges $e_{ij}$ of $G$, where $i$ and $j$ indicate stimuli from the same category (specified by the number of pulses per second for the auditory stimuli or the word categories for the semantic stimuli).

**Inter-categorical similarity.** The mean value of all edges $e_{ij}$ of $G$, where $i$ and $j$ indicate stimuli from different categories in the respective modality.

**Mean clustering coefficient.** This depicts the average clustering coefficient $C$ of $G$: $C = \frac{1}{n}\sum C_u$ where $n$ represents the number of nodes and $c_u$ depicts the clustering coefficient for node $u$ defined by the geometric average of the subgraph edge weights. The clustering coefficient for node $u$ is computed as $C_u = \frac{1}{\deg(u)(\deg(u)-1)}\sum \left(\hat{e}_{uv}\hat{e}_{uw}\hat{e}_{uw}\right)^{1/3}$, where $\hat{e}$ are the normalized edge weights: $\hat{e} = \frac{e_{uv}}{\max(e)}$[48].

**Shortest path length.** This depicts the average shortest path length $A$ of $G$: $A = \frac{1}{n*(n-1)}\sum p_{ij}$ where $n$ is the number of nodes and $p_{ij}$ is the distance of the shortest path between nodes $i$ and $j$ along edges with non-zero weights.

**Global efficiency.** The efficiency $E$ is computed from an unweighted graph $G'$, where edges below a defined threshold (80th percentile of all edge weights) are discarded. The global efficiency $E$ is computed by the average efficiency of all pairs of nodes, where the efficiency of each given pair of nodes is measured by the multiplicative inverse of the shortest path distance between those nodes[49]. Hence, global efficiency quantifies the efficiency of parallel information exchange in a network and can be thought of as an estimate of the small-worldness of a graph. The threshold of the 80th percentile was chosen, after systematically testing different percentiles between 10 and 90, observing qualitative comparability of the resulting efficiency estimates.

**Modularity.** To compute the modularity $Q$ of $G$, the graph $G$ is partitioned into communities according to the categories of stimuli: For the auditory stimulus set, we assumed nine different communities defined by the different numbers of pulses per second – a stimulus feature shown to strongly determine the perception of pulsed sounds[50,51]. For the semantic stimulus set, we assumed five different communities, defined by the word categories. Modularity is then computed as: $Q = \sum\left[\frac{L_c}{m} - \left(\frac{k_c^{in}k_c^{out}}{2m}\right)^2\right]$, where $m$ is the total number of edges, $L_c$ is the number of intra-community links for community $c$, and $k_c$ is the sum of degrees of the nodes in community $c$[52–55].

**Dimensionality.** For this, the covariance matrix $S$ is computed between the centered columns of $M$. The effective dimensionality is then computed by $dimensionality = \frac{\left(\sum w\right)^2}{\sum w^2}$, where $w$ indicates the eigenvalues of the covariance matrix $S$ (found by a singular value decomposition of $S$). The dimensionality thus measures the complexity of the (column-wise) similarity vectors between all stimuli. It can be interpreted as a rough estimate of the needed dimensions to capture the variance between them. A high dimensionality would indicate a more intricate structure of the similarities between the stimuli.

**Correlation analyses and correction for individual biases in the similarity ratings.** In general, we probed the relationships between different individually assessed variables, such as psychometric scores, creativity scores, and individual map features, by computing Pearson correlation. To corroborate the positive correlations, observed for different auditory and semantic map features, we further replicated these positive associations by quantifying Spearman correlation coefficients (see Fig. 2B).

To test if the map features derived from the individual similarity ratings were affected by differences in decision processes, we correlated the average response time during the scaling tasks for each participant with the representational map features in both modalities (see Supplementary Fig. 6). Here, we observed that 14 out of 16 map features did not show a significant correlation to response time, indicating that decision processes only had a minor effect on the representational map estimates in our study.

To control for biases due to the stimulus sets selected in our study, we tested the robustness of the cross-modal correlation of map features against variations in the size of the assessed representational maps. To this end, we randomly subsampled semantic and auditory stimulus sets of predefined size, extracted the corresponding map features for all individuals, and correlated them across modalities ($n = 20$ times random subsampling). Plotting the cross-modal correlation trend of the mean off-diagonal similarity and dimensionality, as representative map features over different subsampling sizes, demonstrates that also smaller estimates of the auditory and semantic map share relational features as indicated by a positive correlation (e.g. already half of the stimuli $n = 10$ suffice to observe significant correlations of $R > 0.15$ for the mean off-diagonal similarity; Supplementary Fig. 3A).

Moreover, we aimed to control the positive association of auditory and semantic map features against individual biases in the similarity ratings, which could theoretically also account for general consistencies of the response patterns, independent from perceptual relations. For instance, a condition where some individuals consistently only use the upper half of the rating scale, combined with other individuals that consistently only use the lower half of the rating scale could also lead to positive correlations across similarity measures. To control for this potential bias, we applied two types of normalization to the raw similarity data: First, we built the z-score of all 210 raw similarity ratings of a participant, normalizing them to the individual dynamic range. Second, we applied a rank-transformation, expressing each similarity rating as the rank in the distribution of all individual 210 ratings. Then, we re-tested the cross-modal correlations of the transformed map features, confirming our initial observation of a positive link between map features across modalities (see Supplementary Fig. 3B, C; the map features that cannot be meaningfully computed from the transformed data are omitted). Together, this indicates that the cross-modal correlation of auditory and semantic map features is robust against subsampling from the stimulus sets and against individual rating biases.

**Multiple linear regression analysis.** To test, if the individual map features in auditory and semantic modality can predict differences in creative abilities, we conducted a multiple linear regression, taking the individual creativity scores as dependent variables and the map features as predictors. To account for correlations between the predictor variables, we applied a ridge regularization. Specifically, we used the 16 individual map features (inter-categorical similarity, mean off-diagonal similarity, mean clustering coefficient, intra-categorical similarity, global efficiency, shortest path length, dimensionality and modularity, each in auditory and semantic modality) to predict each of the four verbal creativity assessments (AUT number of submitted items, AUT semantic distance, C-RAT score, DAT semantic distance). The values for the regularization parameter $\lambda$ were manually chosen after fitting each creativity score with systematically varying $\lambda$-values in the range of 0 to 10 and examining the parameter weights as well as the mean squared error of the models: $\lambda_{AUT\,\#\,items} = 0.02$, $\lambda_{AUT\,sem.dist.} = 0.05$, $\lambda_{CRAT} = 0.05$, $\lambda_{DAT\,sem.dist.} = 0.18$ (different $\lambda$ values did not qualitatively affect the results of our regression analyses). A least-squares algorithm was used for the regressions. To oppose the model goodness of the regression of the empirically assessed creativity scores with randomly permuted control data, we computed corresponding regressions with shuffled map features. Each regression model was cross-validated in a leave-one-out manner, training the model on the data of 118 participants and predicting the data of the remaining one participant. This procedure was repeated for all participants respectively. To compare the predictive power of the map features on the creativity scores, we compared the error distributions over all cross-validations between the real versus shuffled data with a Wilcoxon signed rank test. The values of explained variance ($R^2$) for these models matched the comparison of the prediction errors (Supplementary Table 2).

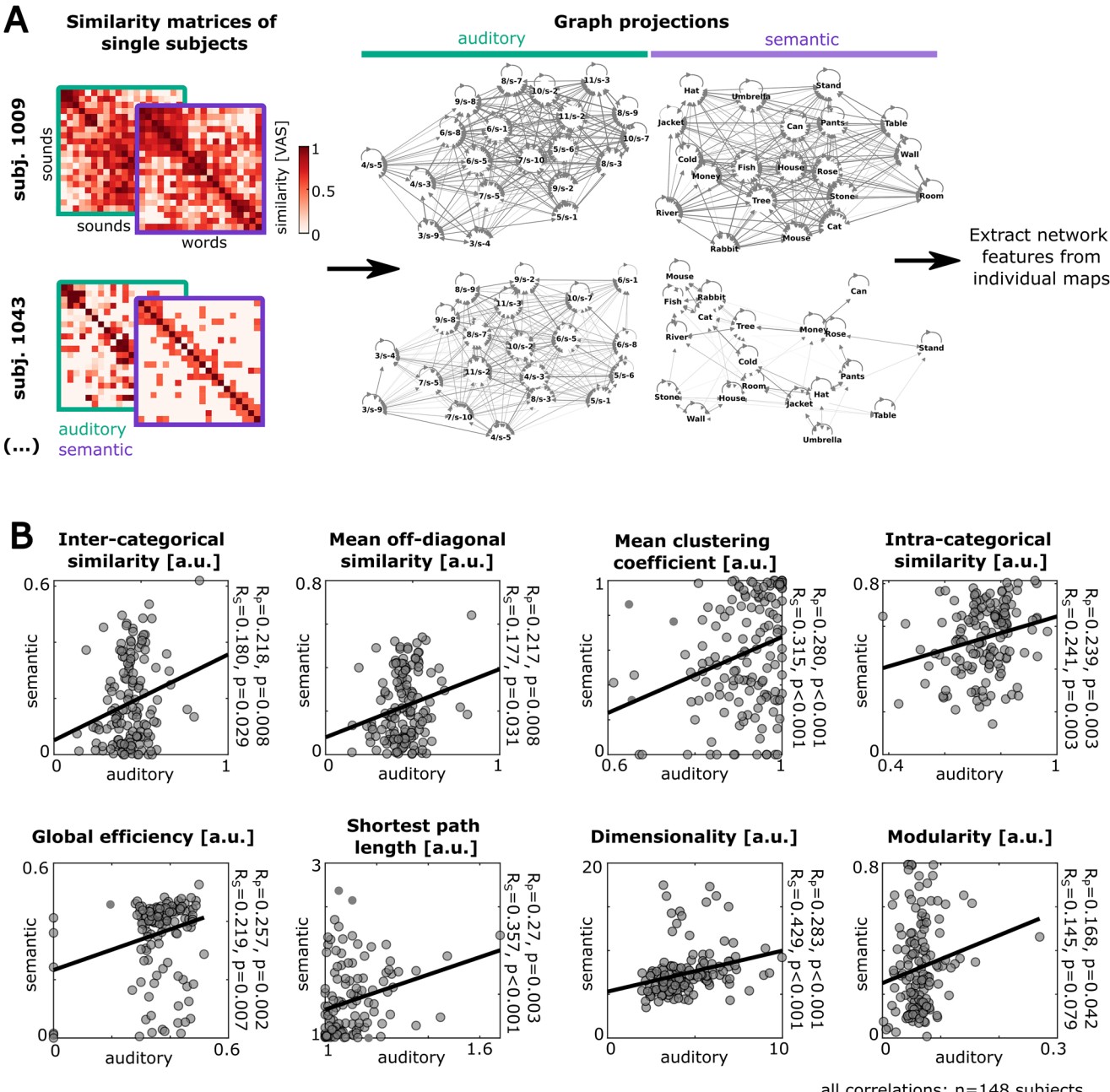

**Fig. 2 | Participants show correlated structural features of their auditory and semantic representational maps.** **A** For all participants ($n = 148$) we estimated the structural features of their auditory and semantic representational map, based on the individual similarity ratings (see "Methods" section): *Inter-categorical similarity*, *mean off-diagonal similarity*, *mean clustering coefficient*, *intra-categorical similarity*, *global efficiency*, *shortest path length*, *dimensionality*, and *modularity*. Exemplary similarity matrices (left) and graph projections (right) in both modalities are shown (for more examples see Supplementary Fig. 2). For the graph, only edges with a weight >0 are displayed. **B** Scatter plots for all map features comparing the auditory against the semantic representational map estimates. Each dot represents an individual participant. The map features robustly show a positive linear ($R_P$) and rank correlation ($R_S$) across modalities.

In order to compare the impact of the different auditory and semantic map features on the prediction of the creativity scores, we computed the contribution of each regression parameter $j$ calculated as: $contribution_j = |x_j \beta_j|$, where $x_j$ denotes the respective map feature and $\beta_j$ indicates the fitted parameter weight.

**Comparing the effects of modality-general vs. modality-specific variance on creativity.** Based on the above regression, expressing creativity scores as a function of individual map features, we sought to compare the role of the *modality-specific* and the *modality-general* fraction of map-feature variance in explaining this relationship. To this end, we used the cross-modal linear correlation of each respective map feature to estimate the shared map-feature variance across modalities: For each participant, we projected the map features on the linear cross-modal fit, using the auditory and semantic axis for the respective projection. Here, the auditory and semantic map feature of the cross-modal projection yields a proxy for the modality-general fraction of the data, whereas the residuals from the linear fit reflect the modality-specific variance in the data. In accordance with our regression using the full map features, we then trained regression models to predict the creativity scores exclusively based on the modality-general or modality-specific map-feature estimations. We compared the error distributions of the modality-general

and modality-specific regression to the regression with the full data using a Wilcoxon signed rank test. We also compared the values of explained variance ($R^2$) for the different regressions, observing congruence with the comparison of error distributions (Supplementary Table 2).

### Reporting summary
Further information on research design is available in the Nature Portfolio Reporting Summary linked to this article.

## Results
### Estimating auditory and semantic representational maps with a perceptual scaling task
The first aim of our study was to assess and compare the representational architectures of an individual in different modalities, covering low-level perceptual features as well as higher-order semantic features, likely relying on different brain areas. Therefore, we conducted a multi-step scaling experiment to assess the representational relations between different stimuli, measured by their relative similarities. We recruited 148 healthy human participants from a pool of university students to participate in our study (see "Methods" section, Supplementary Table 1). Participants first filled out various psychometric questionnaires to generate individual profiles of personality traits and mental health criteria (Fig. 1A, "Methods" section). Next, we assessed each participant's ability to form creative associations, using standard language-based tasks for divergent thinking and creativity ("Methods" section). This was followed by (i) an auditory scaling task and (ii) a semantic scaling task comprising highly dissimilar stimuli (for comparable paradigms used in previous studies see refs. 17,18,56–58): For the auditory task, participants were presented with different pairs of arbitrary pulsed sound stimuli, differing in their temporal pattern of short white noise bursts (see "Methods" section for details). After the presentation of each stimulus pair, the participants were asked to rate the similarity of the two presented sounds without a specified time limit (Fig. 1B). This procedure was repeated to obtain similarity ratings for all pairwise combinations of the 20 sounds in our stimulus set (Fig. 1C). For the semantic task, participants underwent an analogous procedure, rating the similarities in meaning between 20 different word stimuli, presented to them on a computer screen (Fig. 1D, E). For each participant and modality, this task yielded 210 pairwise similarity ratings on a visual analog scale, expressed as numerical values between 0 (no/minimal similarity) and 1 (maximal similarity), which constitute a complex pattern of stimulus relations that can be displayed as a similarity matrix (Fig. 1C, E, see "Methods" section). These similarity matrices provide a simple proxy of individual representational maps for the presented stimuli, assuming that pairs of stimuli with similar internal representation are also perceived and rated alike[59].

We tested the engagement of participants in the rating decisions by comparing the median response times across participants, observing that response times varied within a relatively low range, suggesting largely comparable degrees of task engagement across participants (mean ± SD: 2.47 s ± 0.59 s).

### Participants show coherent patterns of rated similarities
Based on the raw similarity ratings, forming a 20-stimulus by 20-stimulus similarity matrix for each participant and modality, we constructed the mean auditory and semantic similarity matrix across participants, describing the average pairwise similarities between the stimulus pairs of the respective modality (Fig. 1C, E). The matrix for pulsed auditory stimuli showed two main clusters: sounds with 3–4 white noise pulses and sounds containing more pulses. In contrast, the matrix for word stimuli showed a more complex similarity pattern with multiple smaller clusters and higher sparseness in the similarity ratings. The seemingly unrelated patterns in the similarity matrices between the modalities underline the qualitative differences between the used stimulus sets.

To test the reliability of our mean similarity assessments, we randomly subsampled our data to form two evenly-sized groups and computed the correlation of the corresponding mean similarity matrices (see "Methods"

section). Here, both modalities showed reliable similarity profiles with a significant positive correlation over random subsampling (Fig. 1F; $n = 1000$ times random subsampling into two evenly-sized groups, auditory mean corr. ±SD: 0.428 ± 0.014, semantic mean corr. ±SD: 0.401 ± 0.026, $p < 0.001$ in two-sample $t$-test against a mean of zero). In addition, we validated the semantic similarity ratings from our study by comparing them to the similarity patterns of the same word set obtained from a state-of-the-art semantic word embedding model ("Methods" section, Supplementary Fig. 1A, B). Here, the mean semantic similarity ratings widely matched the model reference (Supplementary Fig. 1C; $n = 190$ off-diagonal stimulus pairs, $R_P = 0.298$, $p < 0.001$), indicating consistent structure in our sample's semantic perception.

### Structural features of individual representational maps are shared across the auditory and semantic modalities
We next sought to quantitatively describe the structure of individual auditory and semantic representational maps and test their similarity across modalities. For this purpose, we used the individual raw similarity matrices and constructed corresponding graphs, using the stimuli as nodes and the similarity rating between a given pair of stimuli as edge weight between them (Fig. 2A, see "Methods" section). This graph estimation allowed us to estimate different graph-theoretical properties, referred to as *map features*, that characterize the structure of a representational map in a given participant for the respective stimulus set (Methods). We observed substantial variability in the patterns of the similarity ratings and corresponding graphs across individual participants (Supplementary Fig. 2A, B), indicating a multitude of representational map patterns in the perceptual and semantic domain.

For each participant, we computed map features that primarily describe the connectedness of the graph, i.e. the tendency to rate the stimuli as being perceptually similar (*inter-categorical similarity*, *mean off-diagonal similarity* of the similarity matrix, *intra-categorical similarity*, *global efficiency*, *shortest path length of the graph*), as well as graph features primarily describing the disparity and clusteredness of the graph (*mean clustering coefficient*, *modularity*, *dimensionality*) (see "Methods" section for details about the computation of the map features; see Supplementary Fig. 3A for the correlations between map features). To test for potential confounds explaining the differences in map features, we tested their correlation to the psychometric assessments from our study, finding no significant evidence for interference with personality traits or mental health indicators (Supplementary Fig. 3B; $n = 148$ participants, 72 correlations with $R_P$ in range [−0.167 0.151] and $p$-values in range [0.043 0.944], for a Bonferroni-corrected significance threshold of $6.9*10^{-4}$).

Moreover, as psychophysical responses can potentially be influenced by both, underlying representational structures and executive decision processes, we tested whether the map features estimated for each participant are correlated with the average response time of participants (see "Methods" section). We found that response times as a proxy for decision-related factors only had a minor effect on representational map features (see Supplementary Fig. 6: 14 out of 16 map features did not show a relevant correlation with response times at a Bonferroni-corrected significance level of $p = 0.003$: median response time vs. auditory features: $n = 148$ participants, inter-categorical similarity $R_P = 0.084$, $p = 0.312$/ mean off-diagonal similarity $R_P = 0.087$, $p = 0.295$/ mean clustering coefficient $R_P = 0.154$, $p = 0.061$/ intra-categorical similarity $R_P = 0.136$, $p = 0.100$/ global efficiency $R_P = 0.104$, $p = 0.210$/ shortest path length $R_P = -0.140$, $p = 0.090$/ dimensionality $R_P = -0.252$, $p = 0.002$/ modularity $R_P = -0.039$, $p = 0.643$; response time vs. semantic features: inter-categorical similarity $R_P = 0.142$, $p = 0.086$/ mean off-diagonal similarity $R_P = 0.164$, $p = 0.046$/ mean clustering coefficient $R_P = 0.174$, $p = 0.034$/ intra-categorical similarity $R_P = 0.261$, $p = 0.001$/ global efficiency $R_P = 0.106$, $p = 0.200$/ shortest path length $R_P = -0.144$, $p = 0.120$/ dimensionality $R_P = -0.135$, $p = 0.101$/ modularity $R_P = -0.061$, $p = 0.461$).

In the next step, we tested the consistency of the individual map features between the two probed modalities by computing for each feature the correlation between all auditory and semantic maps across the participants.

Interestingly, we observed significant positive linear and rank correlations across modalities for all of the extracted map features, even despite the fact that the auditory and semantic stimulus sets underlying the map estimation are highly dissimilar (Fig. 2B; $n = 148$ participants). The highest cross-modal correlations were observed for dimensionality ($n = 148$ participants, $R_S = 0.429$, $R_P = 0.283$, $p < 0.001$) and the shortest path length combining two nodes in the graph ($R_S = 0.357$, $R_P = 0.270$, $p < 0.001$).

To control for potential biases due to the size of the map estimates in our study as well as for individual response biases, such as generally higher similarity ratings of a participant, we replicated the positive cross-modal correlation of map features for systematically varying subsamples of the stimulus sets (Supplementary Fig. 4A, "Methods" section; map estimates from: ≥8 stimuli show significant cross-modal correlation of mean off-diagonal similarity over $n = 148$ participants with $p < 0.05$, ≥14 stimuli show significant cross-modal correlation of dimensionality over $n = 148$ participants with $p < 0.05$) and for normalized versions of the similarity ratings evening out individual response biases (Methods; Supplementary Fig. 4B: cross-modal correlation of z-scored map estimates: $n = 148$ participants, inter-categorical similarity $R_P = 0.335$, $p = <0.001$/ mean off-diagonal similarity $R_P = 0.181$, $p = 0.028$/ intra-categorical similarity $R_P = 0.314$, $p = <0.001$/ global efficiency $R_P = 0.257$, $p = 0.002$/ dimensionality $R_P = 0.283$, $p = <0.001$; Supplementary Fig. 4C: cross-modal correlation of rank-transformed map estimates: $n = 148$ participants, inter-categorical similarity $R_P = 0.180$, $p = 0.029$/ mean off-diagonal similarity $R_P = 0.171$, $p = 0.039$/ intra-categorical similarity $R_P = 0.156$, $p = 0.058$/ global efficiency $R_P = 0.247$, $p = 0.003$/ dimensionality $R_P = 0.347$, $p = <0.001$). These control analyses demonstrate the high robustness of the positive cross-modal correlation of auditory and semantic map features. Together, our analyses indicate that features in the representational architecture of an individual are shared across the semantic and perceptual level.

## Predicting creativity scores from individual auditory and semantic map features

Recent studies indicate that the structure and interconnectedness of a person's semantic knowledge network, measured by word association judgments, reflects their creative abilities[17,18]. In these studies, a set of semantic network features were identified as landmarks of an individual's ability to link different pieces of information and thus form innovative associations. Motivated by this observed linkage between creativity and semantic network features, we sought to test if even low-level perceptual maps, assessed by the perception of semantically neutral pulsed sound stimuli, relate to individual creativity. Given the impact of both semantic and perceptual map features on creativity, we further aimed to quantitatively compare the contribution of both modalities.

For this purpose, we first tested the association of the aforementioned auditory and semantic map features from our experiment with standard verbal measures of creativity. Specifically, we probed creative thinking by applying the Alternative Uses Task (AUT), the Compound Remote Association Task (C-RAT), and the Divergent Association Task (DAT), all scored with respect to the semantic variety and fluency of participants' responses (see "Methods" section for details). We first confirmed the validity of the different creativity scores by correlating them with each other, observing robust positive correlations between the creativity measures (Supplementary Fig. 5A; Correlations between creativity measures: $n = 130$ participants, AUT #items vs. AUT sem. dist.: $R_P = 0.627$, $p < 0.001$, AUT #items vs. C-RAT: $R_P = 0.265$, $p = 0.004$, AUT #items vs. DAT: $R_P = 0.075$, $p = 0.432$, AUT sem. dist. vs. C-RAT: $R_P = 0.269$, $p = 0.003$, AUT sem. dist. vs. DAT: $R_P = 0.108$, $p = 0.258$, C-RAT vs. DAT: $R_P = 0.257$, $p = 0.006$). Moreover, in our cohort of healthy participants, we did not find significant evidence for an association of the creativity scores with other, potentially confounding psychometric traits, measured by self-report assessments (Supplementary Fig. 5B; 36 pairwise correlations between creativity scores and psychometric questionnaires: $n = 130$ participants, $p > 0.016$ (Bonferroni-corrected significance threshold)). Testing the correlation of creativity scores and the different representational map features separately, in order to

delineate modalities-specific effects, we observed a trend of associations between map features and creativity scores that was similar for both modalities (Fig. 3A). When testing the consistency of the associations between map features and creativity scores across the two modalities, we found a significant positive correlation (Fig. 3B, $n = 32$ correlations per modality, $R_P = 0.574$, $p < 0.001$), supporting the idea of representational features across modalities being linked to creativity. These findings are consistent with prior studies, observing correlations between semantic network features and creativity[17,18]: Features that reflect high connectivity within a representational map (e.g. intra-categorical similarity) tend to show positive associations with creativity, whereas features that reflect distinctness of elements on a map (e.g. dimensionality) tend to show a negative link to creativity.

Encouraged by these observations, we next tested if the set of auditory and semantic map features in our study could be used to predict individual creativity scores. For this purpose, we trained a linear regression model to predict participants' creativity scores from the pooled auditory and semantic map features simultaneously and tested it using left-out data. In this approach, we intentionally did not average the map features across modalities to allow a delineation of the contributions of each modality independently. We then compared the prediction accuracy of this model with regressions of the shuffled data (Fig. 3C, "Methods" section). We observed a significantly lower error of the regression models trained on the real data, compared to the shuffled condition for all of the creativity scores (Fig. 3D; $n = 119$ participants, two-sided t-test of real vs. shuffled model error: AUT #items: $p = 0.006$, AUT semantic distance: $p < 0.001$, C-RAT #correct: $p = 0.018$, DAT semantic distance: $p < 0.001$; for explained variance see Supplementary Table 2). This indicates that the representational structures of both, perceptual auditory and semantic maps, carry significant information about an individual's potential to form creative associations.

As it was shown that global brain states, such as mood, can affect the structure of representational maps[60,61] and creative performance[62,63], we tested to what extent self-reported mood during the experiment impacted our results. In particular, we evaluated specific mood-related items from our psychometric questionnaires and tested their correlations to the creativity scores and map features of all participants. Here, we did not find significant evidence for an interaction between mood and creativity or the map features in our experiment (Supplementary Fig. 7A; 80 pairwise correlations of self-reported mood and creativity or map features, respectively: $n = 119$ participants, 80 correlations with $R_P$ in the range [−0.138 0.180] and $p$-values in the range [0.028 0.992], for a Bonferroni-corrected significance threshold of 0.008). Incorporating mood-related self-reports into our regression analysis and comparing it to the regression without mood assessments, did not lead to a significant improvement in the prediction of creativity scores (Supplementary Fig. 7B; Two-sided $t$-test of model error for initial regression vs. regression including mood assessments: AUT #items: $p = 0.254$, AUT sem. dist.: $p = 0.713$, C-RAT #correct: $p = 0.248$, DAT sem. dist.: $p = 0.087$).

In addition, we replicated our analyses using an alternative scoring method (OCSAI) to evaluate the originality of AUT responses[47], finding that these scores were significantly correlated with semantic distance (Supplementary Fig. 8A; $n = 118$ participants, $R_P = 0.346$, $p < 0.001$) and also showed significant predictability by participants' representational map features, equivalent to semantic distance (Supplementary Fig. 8B; $n = 119$ participants, two-sided $t$-test of real vs. shuffled model error: $p < 0.001$).

In order to quantitatively compare the impact of the different map features on each creativity score, we computed the parameter contributions from the weights of the regression models (see "Methods" section). Interestingly, not only semantic map features but also features of the auditory representational maps showed a remarkable effect on the prediction of creativity scores (Fig. 3E). Especially, the shortest path length, the mean clustering coefficient, the dimensionality as well as the mean intra-categorical similarity showed high contributions in both modalities (Kruskal–Wallis test indicating significant differences in the overall distributions of parameter contributions: $n = 119$ participants, $p < 0.001$ for all creativity scores). Together, these findings suggest that individual creative

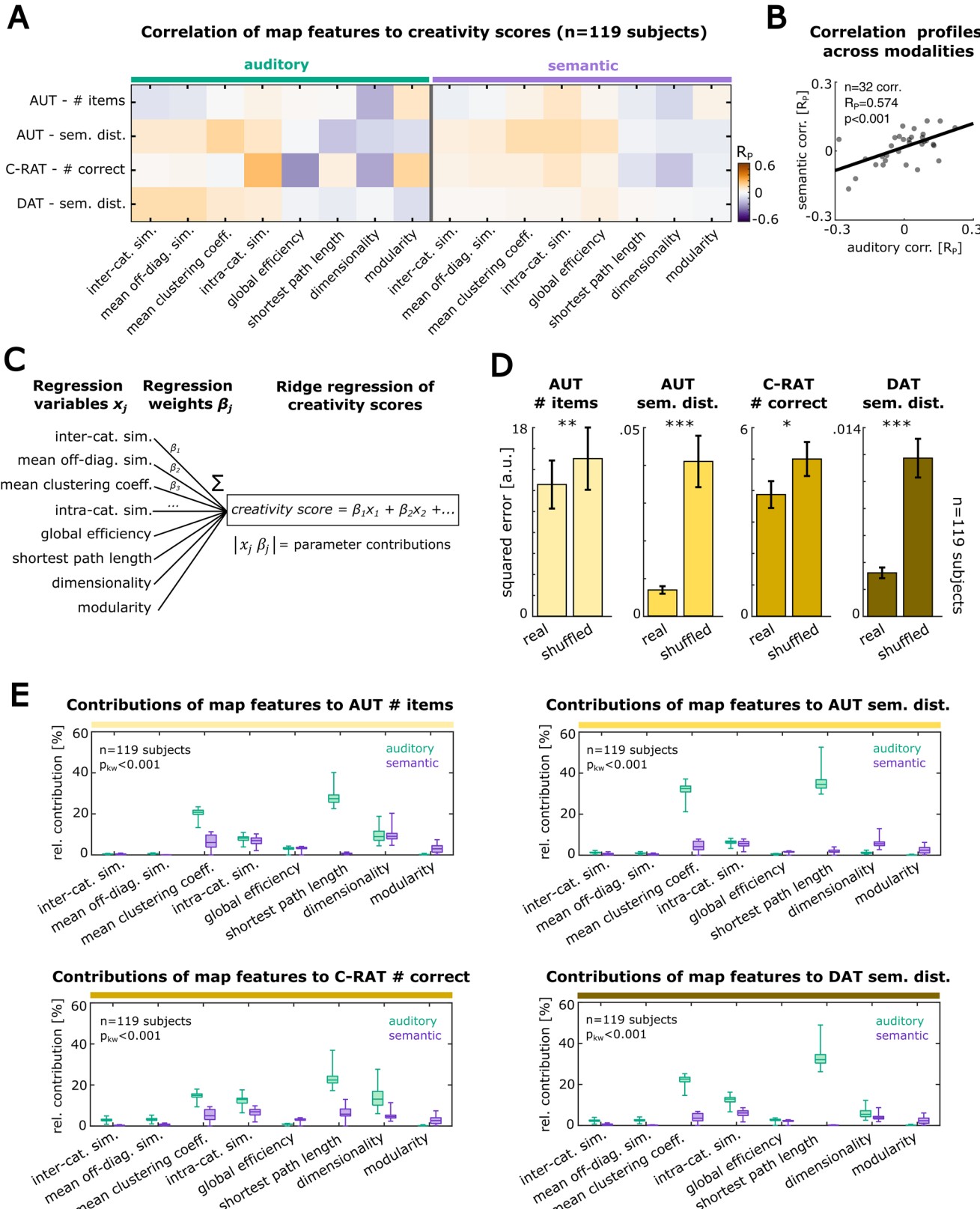

**Fig. 3 | Structural features of auditory and semantic maps predict creativity scores. A** Pearson correlation matrix of the verbal creativity scores and the auditory and semantic map features (*n* = 119 participants). **B** Correlation of the similarity profiles of the auditory versus semantic map features and the creativity scores from (**A**) (*n* = 32 pairwise correlations). **C** To test if auditory and semantic map features can predict individual creativity, we fitted a linear ridge regression model to each creativity score using the pooled map features from both modalities. **D** Testing the goodness of these models against equivalent regression models, trained on the shuffled map features, demonstrates significant predictability for all creativity scores (*n* = 119 participants, \**p* < 0.05, \*\*\**p* < 0.001 in a Wilcoxon signed rank test, see Supplementary Table 2 for explained variances). **E** Comparison of the contributions of all verbal and semantic map features in predicting the creativity scores (see "Methods" section). Except for a few outliers like shortest path length, the contributions of the map features show consistent patterns across modalities.

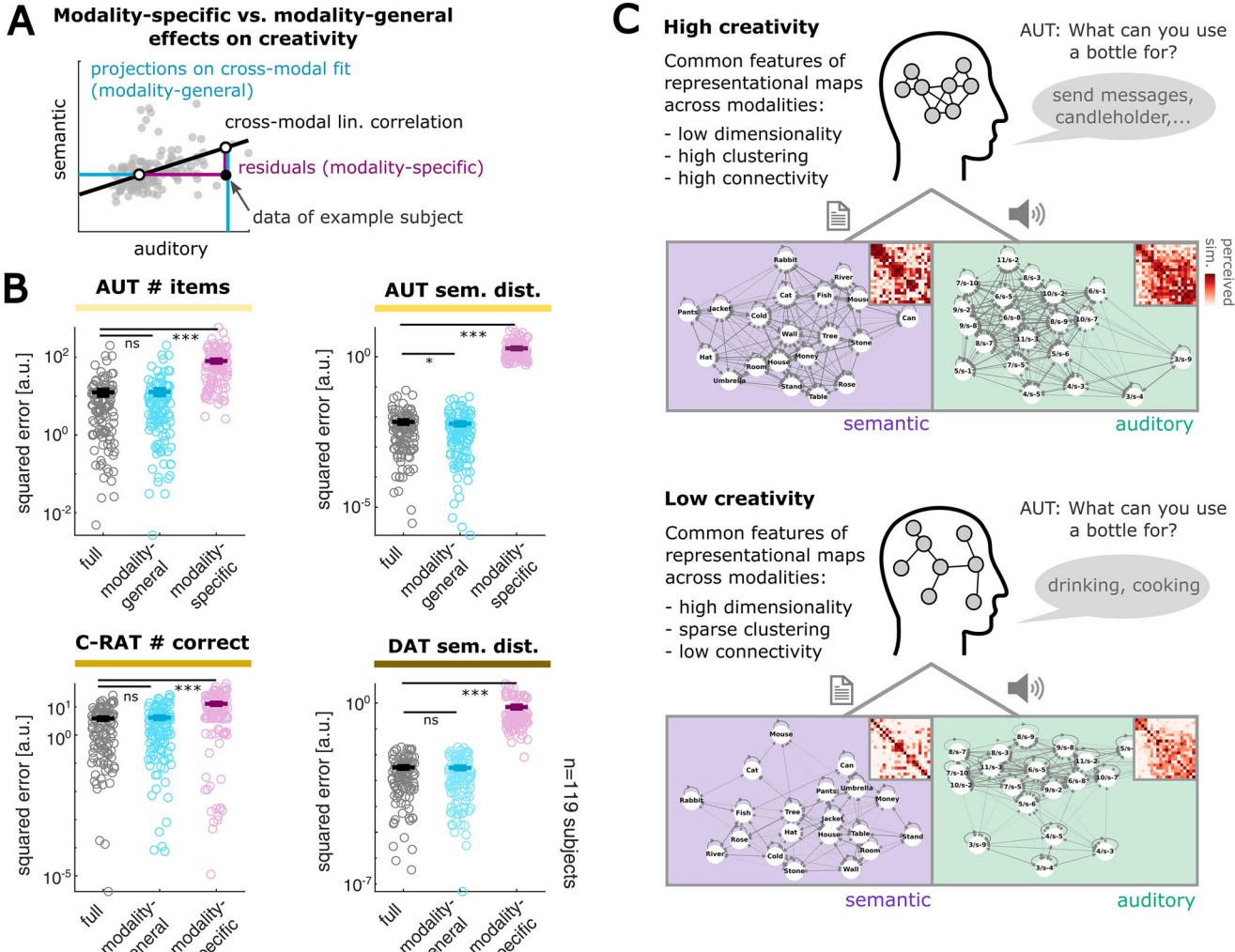

**Fig. 4 | Creative abilities are best predicted by representational features that are general rather than specific for modalities. A** Illustration of how to delineate the *modality-general* and *modality-specific* fraction of variance in map features (see "Methods" section for details). Estimates of modality-general and modality-specific map features were then used for regression analyses of participants' creativity scores. **B** Comparison of the prediction error of the linear regression computed with the full (black, equivalent to errors in Fig. 3D), modality-general (cyan), and modality-specific (magenta) map-feature estimates. Bars indicate the mean values, vertical bars indicate the SEM, and circles represent individual prediction errors (*n* = 119 participants). The modality-general data is a better predictor of all the different creativity scores, as compared to the modality-specific data (*\*p* < 0.05, *\*\*\*p* < 0.001

in a Wilcoxon signed rank test against the initial regression on the raw map features; for explained variance see Supplementary Table 2). **C** Summary schematic illustrating the common representational structures that characterize semantic and auditory maps in individuals with high versus low creativity. The upper and lower panel depict the AUT responses and the similarity ratings from two representative participants with high versus low semantic distance in the AUT. A highly creative participant, with the ability to form unconventional verbal associations, tends to perceive auditory and semantic stimulus percepts as more similar (top). In contrast, a hardly creative participant, forming conventional verbal associations, tends to perceive the stimuli as unrelated and more distinct (bottom).

abilities are reflected by specific structural features of representational maps, not only on a higher semantic level but also on a lower perceptual level.

## Common representational features across modalities explain individual creativity scores

Finally, having observed the effects of both auditory and semantic map features in explaining creativity scores, we sought to test whether this relationship is driven by the *modality-specific* or *modality-general* variance in the map features. To address this question, we specifically estimated the modality-general variance in each map feature, projecting each individual's data on the cross-modal linear fit of this feature (Fig. 4A, "Methods" section). Conversely, we assessed the modality-specific fraction of variance in map features by quantifying each participant's residuals from the linear cross-modal correlation ("Methods" section). This method yields independent estimates of the shared vs. unique fraction of variance in the representational map features for each participant. With

these estimates, we then conducted comparative regressions of the creativity scores, equivalent to our initial analysis. If the relationship between creativity and representational map features should reflect domain-overarching properties, we hypothesized that the modality-general regression should show a lower prediction error than the modality-specific regression. Indeed, we found that the modality-general map features predicted creativity scores significantly better than the modality-specific estimates (Fig. 4B; *n* = 119 participants, Wilcoxon signed rank test: full vs. modality-general / full vs. modality-specific error: AUT #items: *p* = 0.370/ < 0.001, AUT semantic distance: *p* = 0.033/ < 0.001, C-RAT #correct: *p* = 0.139/ < 0.001, DAT semantic distance: *p* = 0.678/ < 0.001; for explained variance see Supplementary Table 2). The semantic distance of the Alternative Uses Task was even better predicted by the modality-general data as compared to the full data of the initial regression (mean squared error ± SEM: full $(6.88 \pm 0.99)*10^{-3}$ vs. modality-general $(5.98 \pm 0.76)*10^{-3}$, Wilcoxon signed rank test:

$p = 0.033$). This suggests that individual creative ability is associated with common features of the auditory and semantic representational architecture, rather than with modality-specific representational features.

Together, our analyses demonstrate that specific structural features of representational maps in an individual are conserved across the auditory and semantic domain and that these shared structural features relate to the capability to form creative associations: Specifically, we observed that high creativity is associated with highly inter-linked perceptual maps, whereas low creativity is associated with sparsely inter-linked perceptual maps (Fig. 4C).

## Discussion

In our study, we employed psychophysics to investigate the representational structures for largely unrelated sets of auditory and semantic stimuli and their links to creativity in a broad cohort of healthy human participants. We quantified individual representational structures of the auditory and the semantic stimuli using graph-theoretical measures, observing that participants exhibit specific representational architectures that are shared between the auditory and semantic modalities. These shared features are robust against corrections for individual response biases and predict the individual ability to form creative associations. Our results indicate that, to a degree larger than previously anticipated, general attributes shape the representational architectures of a person that mediate perceptual and associative abilities across different fields of cognition.

## Limitations

What are potential mechanisms that can explain the correlation of representational map estimates across modalities and their link with creative performance? In our study, we estimate individual representational relations based on psychophysical scaling tasks, demanding participants to rate the similarity between pairs of auditory and semantic stimuli, respectively. While our finding on its own demonstrates the usefulness of psychophysically quantifying representational maps as a proxy for divergent thinking and creativity[40,64], shared structural patterns in the ratings of auditory and semantic similarity can potentially be influenced by multiple, independent factors ranging from the level of incoming stimuli, over individual representational structures and modulators of representational maps, up to executive properties of an individual that generally affect its decisions. In other words: (i) Correlated features of incoming stimuli from different modalities (e.g., the sound of a car and the written word "car") could translate into similar psychophysical ratings, due to the involvement of similar processing pathways. (ii) Shared features in the neuronal implementation of representational maps could lead to correlated similarity ratings across multiple modalities. (iii) Simultaneous modulation of representational maps in the brain by independent higher-order factors, such as intelligence or brain states, could enforce specific structural changes that manifest broadly on representational maps in different modalities. (iv) General readout mechanisms of representational maps might be common across modalities, thus leading to psychophysical similarity by shared decision processes involved in rating different stimuli. In the following, we point-by-point discuss our findings with respect to each of these factors.

## Mechanistic interpretations

(i) In our study design, we intentionally designed the auditory and semantic stimuli to be highly dissimilar. Given the arbitrary nature of the auditory pulsed stimuli and their lack of semantic meaning, we aimed to minimize interference between modalities. This independence of both stimulus sets was reflected by the distinct average similarity patterns of auditory and semantic modalities (see Fig. 1E). With this, the peripheral processing chain for auditory and semantic stimuli in our study should be largely distinct, possibly allowing for independent representations and involvement of different brain structures across modalities[4,65].

(ii) On the level of representational structures, it is an established observation that psychophysical similarity ratings, equivalent to the method used in our study, show high correspondence to similarity patterns between neuronal stimulus representations, assessed via imaging techniques[29,66]. In line with this, previous studies have demonstrated higher connectivity and higher small-worldness in psychophysically estimated semantic networks of highly creative individuals[17,18], suggesting that neural architectures, generally enhancing linkages of activity states across modalities, may account for the unconventional associations formed by highly creative persons[20,67–69].

(iii) Besides the bottom-up influence that stimulus processing has on the structure of a representational map, also top-down influence, such as brain states or other higher cognitive features can affect representational relations[8,60,64,70–73], e.g. by affecting the neuronal activity states accessible to the brain[61,74]. Mood is a prominent example of a brain state, that is known to impact individual creativity[62,63]. In our study, we assessed self-reported mood but did not find significant evidence for an effect of mood on the results of our study (see Supplementary Fig. 7). Moreover, while prior studies have also observed a positive association of creativity with the personality trait of openness[75–77], we did not replicate this effect in our study (see Supplementary Fig. 5B). Other cognitive factors may possibly further contribute to the representational structures that facilitate associations and knowledge retrieval. Prominently, intelligence has been linked with creativity[71,78,79] and was shown to profoundly impact thought trajectories by affecting attention[80] and executive control[81,82]. In our study, we did not control for intelligence, leaving open that intelligence may contribute as a top-down influence to the individual representational architecture to foster creative thought[82].

(iv) Lastly, decision processes that generally affect the readout of perceptual maps in different modalities, could also contribute to similarity ratings as used in our study. To assess the potential influence of general decision processes, we quantified the average response times of participants in the scaling tasks and correlated them to the representational map features derived from the similarity ratings. While very few map features correlated with response time, this effect was comparably small, and most features were uncorrelated to response times (see Supplementary Fig. 6). Furthermore, we conducted specific validation analyses to control the association of creativity and map features against individual rating biases. These control analyses minimize decision-related biases of participants to generally rate higher or lower similarities, by normalizing each individual's responses to its own dynamic range (see Supplementary Fig. 4). By reproducing the cross-modal correlation of representational features, these analyses demonstrate that decision-related variables only have a minor effect on psychophysical map estimates in our study.

Taken together, our analyses of different influencing factors of similarity ratings, indicate that the similarity ratings in our study mostly reflect representational structures and cognitive features, rather than mere executive decision factors or affective states. The dynamical reconfiguration of representational structures in an individual based on its higher cognitive abilities and the current sensory inputs may thus be a mechanism to allow creative associations and the development of efficient solutions in changing environmental conditions.

## Outlook to future validation studies

To further delineate the contribution of different influences on representational maps, future studies could complement psychophysical estimates of stimulus similarities with parallel neurometric assessments of the representational relations in the same participants. It has been shown that neural estimates of representational maps capture various aspects of stimulus categorization[83–87] and associative processes to flexibly adjust and optimize behavioral strategies[2,73]. Thus, a combination of psychophysical and neural assessments could not only be used to confirm that neural representations share structural features across modalities and brain areas, but also decipher the neuronal mechanisms that dynamically update these

representations at a sensory bottom-up level and a higher cognitive top-down level. Moreover, future similarity assessments could characterize stimuli along multiple defined dimensions, such as e.g. rhythmic, phonetic, and semantic similarity, to obtain a more detailed description of their relations. In this context, recent methodological advances with respect to multilayer and multiplex approaches for modeling representational maps and networks will further help to identify how different dimensions of mapped information interact, in order to form complex representational architectures[16]. Our study provides evidence that psychophysically estimated representational maps depict an integrative proxy of multiple, non-exclusive mechanisms, enhancing the formation of creative associations in an individual.

### Relevance for basic neuroscience

Notably, when comparing perceptual auditory and higher-order semantic modalities, we found that not only the semantic map features but also auditory map features were predictive of individual creativity scores. In our dataset, the linkage of creativity to the auditory map features appeared even more strongly than to the semantic ones. However, it should be noted that in the experimental design of our study, the order of the two scaling tasks was fixed and hence potentially introduced a sequence effect. Nevertheless, our findings indicate that besides language-based paradigms, currently building the vast majority of methods in creativity research[20], associative abilities of an individual could also be assessed by behavioral tasks quantifying the perceptual relations between sensory stimuli. This finding is also consistent with the classic idea that certain computational principles may be shared across cortical brain areas and may provide an individual fingerprint across various functional levels of hierarchy[88–90]. Recent observations of shared activity patterns in the early developing cortex[91] could explain the domain-overarching consistency of representational patterns in individuals[91] as well as other cross-modal correlations of cognitive performance[92,93] and the stability of representational architectures despite fluctuating activity at the single-neuron level[94–96].

In this context, our study indicates that individual differences in cortical microcircuits – perhaps due to genetic or environmental predispositions - could likely be shared across different cortical areas, may they be involved in basic perceptual or in higher semantic processing. In this framework, features that serve creativity in the semantic domain would also be reflected when assessing perceptual processing. Moreover, building a perceptual linkage between incoming sensory information might be a crucial factor in many creative processes to enable associations between higher-order representations that relate to this sensory input[97,98], and hence optimize the inferences drawn from a representational network or map. Future studies could identify further perceptual and behavioral characteristics of creative participants, covering additional sensory modalities. Such perceptual tasks for creativity would offer the possibility to be applied even to non-human model organisms, possibly yielding translational insights about basic neurocognitive underpinnings of creativity[99].

Taken together, our study identifies general features of individual representational architectures that are consistent across the auditory and semantic modalities. These features allow a quantitative description of an individual's perception and furthermore correlate with their ability to form creative associations. Thus, creative thinking employs specific patterns of storing information in a relational context, reaching into fundamental processes of perception to optimize behavioral strategies.

### Data availability

The data of this study and the pulsed auditory stimuli used in this study are available via https://doi.org/10.12751/g-node.757evj, or upon request from the corresponding authors.

### Code availability

The MATLAB code used to analyze the data of this study is available via https://doi.org/10.12751/g-node.757evj, or upon request from the corresponding authors.

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

## Acknowledgements
We thank all the participants who volunteered to contribute to this study. Moreover, we thank the team from the Mainz Behavioral and Experimental Laboratory (MABELLA) for the support during the data acquisition. We further thank Jens-Bastian Eppler for the critical and valuable discussions about this study. This work was supported by research grant Deutsche Forschungsgemeinschaft CRC1080-C05 (S.R.), Deutsche Forschungsgemeinschaft SPP 2041 Project #347573108 (S.R.), Deutsche Forschungsgemeinschaft/Agence nationale de la recherche Project #431393205 (S.R.), Deutsche Forschungsgemeinschaft DIP "Neurobiology of Forgetting" (J.E., M.K., S.R.), Rhine-Main University Alliance RMU (S.R., M.K.) and a Focus Translational Neuroscience Mainz fellowship (J.S.). The funders had no role in study design, data collection, and analysis, the decision to publish, or the preparation of the manuscript.

## Author contributions
J.S., J.E., and S.R. designed the study. J.S. requested the study permission from the local ethics committee. A.G. and J.S. recruited the participants and collected the data. J.S. and J.E. analyzed the data. M.K. and S.R. jointly supervised the study with equal contribution. J.S. and J.E. wrote the first draft of the manuscript. All authors edited the manuscript.

## Funding

## Competing interests
The authors declare no competing interests.
