## [Transparent Peer Review file · Communications Psychology]

Perceptual and semantic maps in individual humans share structural features that predict creative abilities

Corresponding Author: Dr Johannes Seiler

Version 0:

Decision Letter: first round

Dear Dr Seiler,

Thank you for your patience during the peer-review process. Your manuscript titled "Perceptual and semantic maps in individual humans share structural features that predict creative abilities" has now been seen by 2 reviewers, and I include their comments at the end of this message. They find your work of interest but raised some important points. We are interested in the possibility of publishing your study in Communications Psychology, but would like to consider your responses to these concerns and assess a revised manuscript before we make a final decision on publication.

We therefore invite you to revise and resubmit your manuscript, along with a point-by-point response to the reviewers. Please highlight all changes in the manuscript text file.

Editorially, we consider it crucial that the revised manuscript addresses Reviewer #2's concerns, acknowledging any ambiguities or caveats and transparently discussing limitations regarding the mechanistic insights offered by the work. Please also make sure that the methodology is clearly explained in the revised manuscript.

I am attaching an Editorial Requests Table that details critical reporting requirements for the revised manuscript. Please attend to each item and ensure your manuscript is fully compliant. If your revised manuscript is not aligned with these requests on major issues, such as those concerning statistics, it may be returned to you for further revisions without re-review.

Please submit the following items:

- Revised manuscript
- Point-by-point response to the referees' comments
- Cover letter (as a separate document)
- <https://www.nature.com/documents/nr-reporting-summary.zip>>Nature Research Reporting Summary
- <https://www.nature.com/documents/nr-editorial-policy-checklist.pdf>>Editorial Policy Checklist
- Completed Editorial Request Table (attached).

via this link: Link Redacted .

Additional guidance is available in our style and formatting guide Communications Psychology formatting guide.

Best regards,

Troy Lui

Troy Lui, PhD
Associate Editor
Communications Psychology

REVIEWER EXPERTISE:

Reviewer #1: creativity, cognitive map

Reviewer #2: cognitive map, network science

REVIEWER REPORTS:

Reviewer #1 (Remarks to the Author):

This study represents a significant advance in our understanding of human creativity. It breaks new methodological ground in quantifying perceptual networks---extending prior work on semantic networks---and shows the unique contribution of such "lower" perceptual systems to high-level creative cognition. The work is rigorous, and the findings are novel.

I reviewed an earlier version of this paper. My previous comments have been addressed and I have nothing further to add.

Reviewer #2 (Remarks to the Author):

Review of commspsych_1273

Perceptual and semantic maps in individual humans share structural features that predict creative abilities

Summary

This manuscript describes a study where similarity ratings of auditory and lexical stimuli are used to create individual-level perceptual and semantic network maps. The key theoretical contribution is to show that network features derived from the maps of both modalities (particularly from a rating task of auditory stimuli with no semantic referent), are associated with creativity scores. The authors argue that this shows that modality-general representational structure underlies creative abilities. Overall, this is an ambitious study with a complex but well-articulated design and the writing is very clear. My opinion is that while the network analyses and control analyses appear to be well conducted, it is not clear to me if the authors' main conclusion can be made so strongly and convincingly. This is expanded upon in the first point of the 'Comments' section below.

Comments

Disentangling the contributors of creative ability: It is well established that creativity is strongly associated with fluid intelligence, and executive cognition such as executive and attentional control, and also well-recognized that creativity is likely a complex by-product of bottom-up and top-down processes (references below). As far as I can tell, these were not measured in the current study and so the contributors of these potential 'top-down' variables in moderating or contributing to the inference of perceptual similarity in the rating tasks cannot be completely ruled out. Of course, it is not possible to obtain these variables from participants given that the study is over, but in the present manuscript it may be too strong of a conclusion to say that creativity is primarily driven by representational differences inferred from behavioral responses that are necessarily an interactive outcome of both perceptual information and top-down cognitive processes operating on those

perceptions, particularly when the design of the study is ultimately correlational in nature. To put it in another way, when a creative participant shows higher similarity ratings across items, is that due mostly to lower-level perceptual processes, or mostly due to a higher-order cognitive process or inference that infers possible sources of similarities across pairs of items? How can the current study results and design distinguish between these possible explanations?

Construction of perceptual and semantic maps, line 542: In this section I wondered if the similarity matrices were constructed based on normalized ratings? It would be good to clarify this early in the manuscript. Does the normalization result in fully connected networks since the edges are presumably present in most comparisons and the edge weights are continuously scaled? This does not seem to be the case based on the example networks in the supplementary materials. Generally, I was not able to have a clear enough picture of how the actual raw behavioral data was converted into the networks. Perhaps a concrete walkthrough or example would be helpful here.

Other minor comments about the methods:

line 563 80th percentile as a cut-off seemed rather arbitrary, could the authors explain their rationale?

line 569 the computation of modularity seems rather different from the literature on semantic networks and creativity (referencing the body of work by Kenett and colleagues). Usually the graph is submitted to a community detection algorithm rather than imposing the communities based on categorical labels of the nodes/items. Could the authors explain their rationale for their modularity computation further?

Data availability statement: Given current norms about open science and data, it would be difficult for me to recommend publication unless the authors make their stimuli, data, analysis scripts freely available to the scientific community. From experience, data requests are not readily honored by publishing authors.

References

Nusbaum, E. C., & Silvia, P. J. (2011). Are intelligence and creativity really so different?: Fluid intelligence, executive processes, and strategy use in divergent thinking. *Intelligence*, 39(1), 36-45.

Frith, E., Kane, M. J., Welhaf, M. S., Christensen, A. P., Silvia, P. J., & Beaty, R. E. (2021). Keeping creativity under control: Contributions of attention control and fluid intelligence to divergent thinking. *Creativity Research Journal*, 33(2), 138-157.

Kenett, Y. N., Beaty, R. E., Silvia, P. J., Anaki, D., & Faust, M. (2016). Structure and flexibility: Investigating the relation between the structure of the mental lexicon, fluid intelligence, and creative achievement. *Psychology of Aesthetics, Creativity, and the Arts*, 10(4), 377.

* TRANSPARENT PEER REVIEW: Communications Psychology uses a transparent peer review system. This means that we publish the editorial decision letters including Reviewers' comments to the authors and the author rebuttal letters online as a supplementary peer review file. However, on author request, confidential information and data can be removed from the published reviewer reports and rebuttal letters prior to publication. If your manuscript has been previously reviewed at another journal, those Reviewers' comments would not form part of the published peer review file.

If you experience problems in linking your ORCID, please contact the Platform Support Helpdesk.

Version 1:

Decision Letter: second round

Dear Dr Seiler,

Your manuscript titled "Perceptual and semantic maps in individual humans share structural features that predict creative abilities" has now been seen by our reviewers, whose comments appear below. In light of their advice I am delighted to say that we are happy, in principle, to publish a suitably revised version in Communications Psychology.

We therefore invite you to revise your paper one last time to address the remaining concerns of our reviewers and a list of editorial requests. At the same time we ask that you edit your manuscript to comply with our format requirements and to maximise the accessibility and therefore the impact of your work.

EDITORIAL REQUESTS:

In particular, we emphasize that the reporting and interpretation of non-significant results in NHST analyses does not yet comply with our policy (<https://www.nature.com/commspsychol/submit/submission-guidelines#statistical-guidelines>). Please ensure that all statistics are reported for non-significant findings just as for significant findings; that null findings obtained in NHST are not presented as positive evidence for the absence of a difference or effect (and not interpreted in the Discussion); and that your interpretation of Bayes Factors is aligned with journal standards. Please avoid making claims based on anecdotal evidence following the convention detailed in Schönbrodt, F.D., Wagenmakers, E. Bayes factor design analysis: Planning for compelling evidence. *Psychon Bull Rev* 25, 128–142 (2018) <https://rdcu.be/b6uOC>

SUBMISSION INFORMATION:

OPEN ACCESS:

* DATA AVAILABILITY:

Link Redacted

Best regards,

Troyby Lui

Troyby Lui, PhD
Associate Editor
Communications Psychology

REVIEWERS' COMMENTS:

Reviewer #1 (Remarks to the Author):

As the reviewer who reviewed this work previously (and favorably), I have no further comments and continue to support the publication of this work.

Reviewer #2 (Remarks to the Author):

I was a previous reviewer of this manuscript and am happy to say that the revisions have adequately addressed my previous comments.

To the
Editorial Board of
Communications Psychology

Research Group for Systemic Neurophysiology

Dr. med. Johannes P.H. Seiler
Group of Prof. S. Rumpel for Systemic Neurophysiology
Hanns-Dieter-Hüsch-Weg 19
55128 Mainz, Germany
Phone: +49 (0) 6131 39-27356
Fax: +49 (0) 6131 39-26071
www.unimedizin-mainz.de/physiologie/ag-prof-simon-rumpel

Mainz, 18/01/2025

Point-by-point response to reviewers' comments on the manuscript "*Perceptual and semantic maps in individual humans share structural features that predict creative abilities*"

All authors would like to thank the reviewers for their time and constructive criticism. These comments have led to adjustments of the manuscript, in our opinion, significantly improving its quality. The respective adjustments with regards to the content of the manuscript are marked with yellow background color in the attached document.

In the following, we address the reviews point by point:

Reviewer #1:

This study represents a significant advance in our understanding of human creativity. It breaks new methodological ground in quantifying perceptual networks---extending prior work on semantic networks---and shows the unique contribution of such "lower" perceptual systems to high-level creative cognition. The work is rigorous, and the findings are novel.

I reviewed an earlier version of this paper. My previous comments have been addressed and I have nothing further to add.

Thank you.

Reviewer #2:

Summary

This manuscript describes a study where similarity ratings of auditory and lexical stimuli are used to create individual-level perceptual and semantic network maps. The key theoretical contribution is to show that network features derived from the maps of both modalities (particularly from a rating task of auditory stimuli with no semantic referent), are associated with creativity scores. The authors argue that this shows that modality-general representational structure underlies creative abilities. Overall, this is an ambitious study with a complex but well-articulated design and the writing is very clear.

Thank you for this positive comment.

My opinion is that while the network analyses and control analyses appear to be well conducted, it is not clear to me if the authors' main conclusion can be made so strongly and convincingly. This is expanded upon in the first point of the 'Comments' section below.

Comments

Disentangling the contributors of creative ability: It is well established that creativity is strongly associated with fluid intelligence, and executive cognition such as executive and attentional control, and also well-recognized that creativity is likely a complex by-product of bottom-up and top-down processes (references below). As far as I can tell, these were not measured in the current study and so the contributors of these potential 'top-down' variables in moderating or contributing to the inference of perceptual similarity in the rating tasks cannot be completely ruled out. Of course, it is not possible to obtain these variables from participants given that the study is over, but in the present manuscript it may be too strong of a conclusion to say that creativity is primarily driven by representational differences inferred from behavioral responses that are necessarily an interactive outcome of both perceptual information and top-down cognitive processes operating on those perceptions, particularly when the design of the study is ultimately correlational in nature. To put it in another way, when a creative participant shows higher similarity ratings across items, is that due mostly to lower-level perceptual processes, or mostly due to a higher-order cognitive process or inference that infers possible sources of similarities across pairs of items? How can the current study results and design distinguish between these possible explanations?

We thank the reviewer for raising this important aspect. We agree that creativity can be affected by various complex higher-order factors, such as intelligence or executive functions. Although, we did not explicitly control for the impact of intelligence or other 'top-down' factors in our study, we try to acknowledge these factors in the Discussion section and we believe that our interpretation is not conflicting with the fact that these higher-order factors can influence creativity.

In our Discussion section, we list different factors that could theoretically impact on the psychometric similarity ratings in our study (l. 571 ff.). Here, we try to emphasize that both – 'top-down' cognitive factors as well as lower-order representational features – can potentially contribute to differences in the psychometric ratings of similarity in our study. Specifically, while we do not find that our data is significantly affected by mood or personality traits (see Supplementary Figure 5B+7), we try to acknowledge that intelligence, as a higher-order cognitive feature, could indeed affect the representational structures of an individual (Noda et al., 2024). It is a discussion in the field how much internal representational structures on the one side and higher-order factors, on the other, are contributing to the judgement of similarities (Beaty and Kenett, 2023). But, importantly, any potential impact of intelligence or other higher cognitive factors on representational structure would not change the interpretation that these representational structures are linked to creative abilities, rather than mere down-stream decision processes.

Moreover, we tested our analyses against confounding by decision-related factors, where we did not find significant confounds (see l. 443 ff., Supplementary Figure 4B+C).

In this context, we would also like to point out work from Kriegeskorte and colleagues, showing that neuronal measures of representational structures of passively presented stimuli assessed by fMRI are in high accordance with active, psychophysical assessments of perceived similarities between stimuli (Mur et al., 2013). We think, this demonstrates that psychophysical assessments, albeit being influenced by higher-order cognitive factors, nevertheless allow a reasonable readout of representational structures themselves.

This led us to the interpretation that the creativity-related similarity ratings in our study are most likely related to differences in the representational architecture of individuals. As the reviewer correctly points out, given our study design, we cannot ensure a causal relationship.

In sum, we are thankful for the reviewer's comment highlighting the complexity in interpreting our psychophysical observations.

To address the reviewer's point, and to better reflect this complexity and make our line of reasoning clearer, we revised the Discussion section, including the mentioned references, and emphasizing explicitly that other higher-order cognitive factors may also affect the representational structures (l. 608 ff.). Moreover, to address this issue in full depth, future studies could combine psychometric similarity assessments with neurometric assessments of representational similarities to check for consistency in both measures. We added this point to the Discussion section (see l. 636 ff.).

To furthermore make explicit that the nature of our study is correlational rather than causal, we changed the sentence, "...similarity ratings in our study are primarily driven by underlying representational and cognitive structures ..." to "...similarity ratings in our study mostly reflect representational structures and cognitive features ..." (see l. 628 ff.).

Construction of perceptual and semantic maps, line 542: In this section I wondered if the similarity matrices were constructed based on normalized ratings? It would be good to clarify this early in the manuscript. Does the normalization result in fully connected networks since the edges are presumably present in most comparisons and the edge weights are continuously scaled? This does not seem to be the case based on the example networks in the supplementary materials.

For the similarity matrices and graphs shown in Figure 1 and 2, we constructed the similarity matrices for each participant from the raw similarity ratings (0-100% similarity rated on a visual analog scale, see below) without applying a normalization. This procedure results in fully connected graph estimates for each similarity matrix, carrying edge weights that cover a continuous range between 0 and 1. Please note that indeed participants commonly rated pairwise similarities with exactly 0. For the visual display of exemplary graphs (e.g. in Figure 2, Suppl. Figure 2), we displayed all edges with a weight larger than zero. A normalization of each participant's similarity matrix (z-scoring, rank-transformation) was only applied in addition when controlling our findings for influences due to individual choice biases (see l. 296 ff., 443 ff. and Supplementary Figure 4).

To make these points clear in the manuscript, we revised and expanded parts of the Methods section and figure legends to elaborate on this information (see l. 233 ff., l. 969).

Generally, I was not able to have a clear enough picture of how the actual raw behavioral data was converted into the networks. Perhaps a concrete walkthrough or example would be helpful here.

We apologize, we did not make our procedure clear enough. To address this point, we adjusted the main text, elaborating on the transformation participants' raw similarity ratings into similarity matrices and corresponding graphs (see l. 229 ff., 381 f., 403 ff.).

Other minor comments about the methods:

line 563 80th percentile as a cut-off seemed rather arbitrary, could the authors explain their rationale?

For the computation of global efficiency, we initially tested different threshold values, ranging between the 10th and 90th percentile of a given graph's edge weights. Despite this wide range of thresholds we obtained qualitatively comparable efficiency estimates, showing an similar correlations across the semantic and auditory domain (correlations ranging between $R_{\text{thresh}=10\%}=0.266$ and $R_{\text{thresh}=90\%}=0.192$). For our study, we chose the 80th percentile, as a rather strict compromise in the tested range of thresholds. We added this information to the Methods section of the revised manuscript (l. 260 ff.).

line 569 the computation of modularity seems rather different from the literature on semantic networks and creativity (referencing the body of work by Kenett and colleagues). Usually the graph is submitted to a community detection algorithm rather than imposing the communities based on categorical labels of the nodes/items. Could the authors explain their rationale for their modularity computation further?

Thank you for this comment. We want to point out that finding the 'best' community partitioning of a graph is a non-trivial problem (Reichardt and Bornholdt, 2006, He et al., 2023). In the scope of our study, the primary question was not to estimate in how many communities a given individual map would be optimally separated, we rather were interested in obtaining a robust measure how easy (or not) the structure of the two maps obtained from an individual could be captured by a community structure. To increase the robustness, we assumed a community structure that reflected the way we had selected and assembled our stimulus sets in the semantic and perceptual tasks: We used carefully selected stimulus sets of noun words and pulsed auditory sounds, respectively. These stimulus sets were designed in a way, that the single stimuli (nodes) can be classified into distinct and hardly overlapping semantic clusters (e.g. animals, clothing, etc.; similar to the stimuli used in (Benedek et al., 2017)), or into clusters with different numbers of sound pulses – a stimulus feature which previously has been shown to guide perception of pulsed sounds (Brunton et al., 2013, Sheppard et al., 2013).

As modularity is defined as a measure of how strongly a network is partitioned into communities, our approach aimed to detect how well the individual networks of participants would be clustered into the assumed common modular structure shared across all participants. Thus, our approach enabled us to quantify the individual deviation of a given participant's network from the imposed community structure. To better describe this rationale in the main text, we adjusted the Methods section of our manuscript (see l. 264 ff.).

Data availability statement: Given current norms about open science and data, it would be difficult for me to recommend publication unless the authors make their stimuli, data, analysis scripts freely available to the scientific community. From experience, data requests are not readily honored by publishing authors.

We agree with the reviewer about the relevance of open science and free access to the data of scientific studies. To address this comment, we uploaded the data and the stimuli of the study as well as the code to analyze it to the following repository: <https://doi.org/10.12751/g-node.757evj>, adding a reference to this link to the data and code availability statements (l. 685 ff.).

References

Nusbaum, E. C., & Silvia, P. J. (2011). Are intelligence and creativity really so different?: Fluid intelligence, executive processes, and strategy use in divergent thinking. *Intelligence, 39*(1), 36-45.

Frith, E., Kane, M. J., Welhaf, M. S., Christensen, A. P., Silvia, P. J., & Beaty, R. E. (2021). Keeping creativity under control: Contributions of attention control and fluid intelligence to divergent thinking. *Creativity Research Journal, 33*(2), 138-157.

Kenett, Y. N., Beaty, R. E., Silvia, P. J., Anaki, D., & Faust, M. (2016). Structure and flexibility: Investigating the relation between the structure of the mental lexicon, fluid intelligence, and creative achievement. *Psychology of Aesthetics, Creativity, and the Arts, 10*(4), 377.

Thank you for pointing out these studies. We have added references to the text of our manuscript, when discussing potential links of creativity and similarity assessments with fluid intelligence (l. 613 ff.).

We further moved the Methods section to be located prior to the Results section of the manuscript. In addition, we added subheadings to structure the discussion section of the manuscript, in order to adhere to the journal guidelines.

Together, we hope that all these adjustments clarify the scientific contribution of our work, setting it into a meaningful context with prior research.

Sincerely,

Johannes Seiler (in the name of all authors)

References:

BEATY, R. E. & KENETT, Y. N. 2023. Associative thinking at the core of creativity. *Trends in Cognitive Sciences, 27*, 671-683.

BENEDEK, M., KENETT, Y. N., UMDASCH, K., ANAKI, D., FAUST, M. & NEUBAUER, A. C. 2017. How semantic memory structure and intelligence contribute to creative thought: a network science approach. *Thinking & Reasoning, 23*, 158-183.

BRUNTON, B. W., BOTVINICK, M. M. & BRODY, C. D. 2013. Rats and Humans Can Optimally Accumulate Evidence for Decision-Making. *Science, 340*, 95-98.

HE, Z., WEI, X., CHEN, W. & LIU, Y. 2023. On the Statistical Significance of a Community Structure. *IEEE Transactions on Knowledge and Data Engineering, 35*, 2887-2900.

MUR, M., MEYS, M., BODURKA, J., GOEBEL, R., BANDETTINI, P. A. & KRIEGESKORTE, N. 2013. Human Object-Similarity Judgments Reflect and Transcend the Primate-IT Object Representation. *Frontiers in Psychology, 4*.

NODA, T., ASCHAUER, D. F., CHAMBERS, A. R., SEILER, J. P. H. & RUMPEL, S. 2024. Representational maps in the brain: concepts, approaches, and applications. *Frontiers in Cellular Neuroscience, 18*.

REICHARDT, J. & BORNHOLDT, S. 2006. When are networks truly modular? *Physica D: Nonlinear Phenomena, 224*, 20-26.

SHEPPARD, J. P., RAPOSO, D. & CHURCHLAND, A. K. 2013. Dynamic weighting of multisensory stimuli shapes decision-making in rats and humans. *Journal of Vision, 13*, 4-4.